# Vision Impairments in Young Adults with Down Syndrome

**Yolanda Martin-Perez \*, Guadalupe Gonzalez-Montero, Angel L. Gutierrez-Hernandez, Vanesa Blázquez-Sánchez and Celia Sánchez-Ramos** 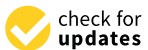

Faculty of Optic and Optometry, Department of Optometry and Vision Science, Complutense University of Madrid, 28040 Madrid, Spain; mggonzalez@ucm.es (G.G.-M.); aguther@ucm.es (A.L.G.-H.); vblazquez@ucm.es (V.B.-S.); celiasr@ucm.es (C.S.-R.)

**\*** Correspondence: ymartinp@ucm.es

**Abstract:** People with Down syndrome have more visual problems than the general population. They experience premature ageing, and they are expected to also have an acceleration in worsening visual function. A prospective observational study which includes visual acuity, refractive error, accommodation, binocular and colour vision was performed on young adults with ($n = 69$) and without ($n = 65$) Down syndrome and on a senior group ($n = 55$) without Down syndrome. Results showed significant differences in visual acuity between groups ($p < 0.001$), and it can be improved with a new prescription in 40% of the participants with Down syndrome. Regarding the accommodative state, no significant differences were found between groups of young people. Concerning binocular vision, 64.7% of strabismus was observed in the group with Down syndrome ($p < 0.001$). Visual abnormalities are significant in young adults with Down syndrome and are different from those of older people without Down syndrome, some of which can be improved by providing the optimal prescription as well as regular eye examinations.

**Keywords:** ageing; Down syndrome; quality of life; refractive error; visual impairments



## 1. Introduction

Down syndrome (DS) is one of the most regular genetic disorders and the most common genetic cause of intellectual disability overall. The estimated incidence of Down syndrome is between 1 in 1000 and 1.4 in 1000 live births in European countries and the United States, respectively [1]. The number of people with DS living in Spain is around 35,000 [2].

DS varies in severity among individuals, and it commonly has other medical conditions associated with cardiac, metabolic, gastrointestinal and respiratory systems [3,4]. Life expectancy has dramatically increased in people with DS due to the medical advances in diagnosis and treatment of the pathologies they present. Currently, in developed countries, people with DS can live for more than 70 years [3,5–7]. Preventive health care is not as frequent as it should be in adolescents and young adults with DS [8]. There is a widespread interest in providing them with the necessary training to enable them to work and be as independent as possible. The more autonomy they have, the better their integration will be.

Nowadays, people with DS have a very active life, so visual demands are greater than they were some years ago. Signs such as redness or watering eyes, squinting, tilting head extremely, placing head too close to the desk while working at near distance and poor attention during near tasks can be directly observable by relatives, heads and teachers. However, there are some symptoms such as blurry or double vision and headaches that people with DS frequently are not able to report because of their communication problems [9–11]. All these conditions are known to be related to visual impairments.

Most of the literature reveals more prevalence in visual and ocular anomalies such as refractive errors, binocular disorders, nystagmus and lack of accommodation in children with DS [12–20], but there is less information about young adults and seniors [20–24].

Visual problems in DS people vary due to the sample size, ethnicity and method of evaluation and may occur in 96% [25]. Reduced VA can be present in 43.3% [21], hyperopia in 46.5%–81.2%, strabismus in between 18.1% and 57.0% and nystagmus may affect up to 33% of individuals with DS [26].

There are recommendations for the timing of basic medical referrals in newborn babies with DS, and these include vision. The first check-up would take place just after birth, and then in the first year every 6 months, to rule out congenital cataracts, glaucoma, nystagmus and retinal pathologies. Later in childhood, visual examinations should be at least once per year or every two years [18]. In childhood, more frequent ocular anomalies detected are strabismus and refractive errors, which is why it is so important to treat them as soon as possible. For adults, the recommendation for visual examinations is every five years [25].

People with DS and other conditions with neurological disorders experience more difficulties in receiving health care than the general population [26,27], and this includes vision care. Sometimes communication problems and mental capabilities make it difficult to apply the same methods used to evaluate vision as is performed in general population. So, that is why eye experts must know how to adapt procedures to get reliable responses to obtain a diagnosis and provide an appropriate treatment [9,11].

Eye problems change with age—for instance, cataracts and presbyopia—which is why regular eye checkups are so important. It has been reported that adults and seniors with DS wear the same glasses that were prescribed when they were children, or the glasses they use are not well adjusted. This fact would explain the lack of compliance in their use [28–30]. Therefore, a prospective study on ocular manifestations in a group of young adults with DS was undertaken and compared to visual impairments in young adults without DS and to seniors without DS. There is a lack of information concerning vision in young people with DS. The size of the sample, the range of age of the group and the method of evaluation provide important information about the vision impairments in DS.

## 2. Materials and Methods

A sample of 69 young adults with DS (Group 1) were recruited from Aprocor Foundation. The aim of this Foundation is to improve the quality of life of people with intellectual disabilities and their families while promoting a model of an inclusive society. The possibility to participate in this research was offered to the head of Aprocor Foundation, and the families which agreed and complied with the inclusion criteria were accepted into the study. The group was recruited from among the people attending Aprocor to perform different tasks and jobs after finishing their school education. People with different disabilities can attend when they are over 17 years of age. People belonging to Aprocor perform different tasks such as painting, handicrafts, packaging, gardening, working with computers or laundry. All the activities are adapted to their capabilities. The typical controls were formed by students from the Optic and Optometry Faculty of the Complutense University in Madrid and were subdivided into two subgroups. Group 2 consisted of 65 young adults (undergraduate students) without DS, and Group 3 consisted of 55 seniors also without DS (students enrolled in a special programme for seniors). Inclusion criteria for Group 1 included DS diagnosed and aged between 17 and 35; for Group 2, the criteria were no DS diagnosed and same age, and for Group 3, no DS diagnosed but aged between 55 and 80.

The study is a prospective observational study. This was explained to all the patients, parents or legal guardians. They agreed to participate through written informed consent, which was signed before the beginning of the study. No drug was instilled, and no invasive technique was performed.

We explained in detail the task they had to perform in each procedure, encouraging them that if they thought they had made a mistake, they could repeat the answer. We did not want them to think that their answer should be immediate or that they had to give a response quickly. We did not want them to find the task frustrating. No specific time was set for each test, and breaks were taken when necessary.

The tests performed on the subjects involved in the study included a visual acuity (VA) test in a high-contrast photopic condition, which was assessed using the Lea symbols 15-line distance chart (Good-Lite, Elgin, IL, USA) at 3 m and Lea symbols near vision card (Good-Lite, Elgin, IL, USA) at 40 cm. These tests were performed on all participants to avoid differences in the interpretation of the results and in order to be more consistent with the statistical analysis. Preliminarily, a trial was carried out to confirm all the participants understood what they had to answer. When participants could not verbalize the answer, they pointed at the symbol in a hand card as it is recommended in the instructions of the test. This measurement was taken with the habitual prescription—with or without glasses—and with the optimal one, after the refractive exam. Measurements were taken monocularly and binocularly and recorded in logMAR units (logarithm of the minimum angle of resolution). For the analysis of the VA, only binocular values were considered.

The current prescription was measured with a lensometer (LM-6 Topcon Co., Tokyo, Japan), and the refractive error was assessed using an autorefractor (TR4000 Tomey, Nürnberg, Germany), a retinoscope (18100 Welch Allyn, Chicago, IL, USA) and retinoscopy rulers. After taking two objective measurements of the refractive error, it was confirmed with a subjective exam using trial frame and loose lenses. The optimal prescription was registered in sphero-cylindrical form, and the lenses that provided the best VA were prescribed. The spherical equivalent (SE) was calculated by adding the sum of the sphere power with half of the cylinder power.

Also, monocular accommodative response was assessed using the push-down method with Lea symbol cards. All the participants wore the optimal prescription for near distance. A card with a single symbol was presented at 4 cm opposite one eye and then moved away until the participant could identify it. The procedure was repeated three times with different cards in each eye. The distance where the symbols were identified was registered in centimetres, and the average of the three measurements was calculated and used as the accommodative response.

Binocular vision was assessed using the cover test method, applied to evaluate the ocular alignment using an occluder, an accommodative Lea symbol test for long distance (3 m) and a single Lea symbol target for near distance (40 cm). To measure the deviation, prism bars were used. It was registered if there were phoria or tropia, the amount, the direction, the constancy and the eye involved. Near point of convergence (NPC) was recorded with the help of a ruler and a card with a single Lea symbol. The break and recovery point were registered in centimetres. The level of stereoacuity was measured for near distance using the Titmus Estereopsis Test (Stereo Optical Co., Chicago, IL, USA). All the measurements were performed wearing the optimal prescription.

Finally, colour vision was analysed, wearing the optimal prescription, with Ishihara's Tests for Colour Deficiency 38 Plates Edition 2000 (Kanehara & Co., Ltd., Tokio, Japan) and Album Tritan de Ph. Lanthony (Laboratoire de la vision des couleurs, Centre National d'Ophalmologie des Quinze Vingts, Paris, France).

For the statistical analysis, all the data were exported to the SPSS 22 version for Windows (Inc., Chicago, IL, USA). The data are presented as the mean $\pm$ standard deviation (SD). The measured outcome variables were checked for normality with the Kolmogórov–Smirnov test. The Wilcoxon test showed that there was no statistically significant difference between the right and left eyes in the refractive error. For no parametric variables, Kruskal–Wallis analysis for independent samples was used. For results obtained in the same subjects, the Greenhouse–Geisser test was applied. Those variables with a p-value of less than 0.05 were considered to have a statistically significant association.

## 3. Results

The total sample was formed of 192 subjects: 69 young adults with DS (Group 1) and a control group without DS. The control group was subdivided into two other groups: 68 young adults (Group 2) and 55 senior adults (Group 3). The characteristics of the participants are in Table 1.

**Table 1.** Demographics of the participants.

| | Groups | *n* | Age (Years) Range | Mean Age (SD) | Gender | |
| | | | | | Male *n* (%) | Female *n* (%) |
|---|---|---|---|---|---|---|
| Subjects with Down syndrome | Group 1 | 69 | 17–35 | 27.6 ± 3.9 | 24 (34.8) | 45 (65.2) |
| Subjects without Down syndrome (control) | Group 2 | 68 | 19–34 | 22.4 ± 2.6 | 17 (25) | 51 (75) |
| | Group 3 | 55 | 55–79 | 63.4 ± 5.7 | 26 (47.2) | 29 (52.8) |

Group 1: young adults with Down syndrome; Group 2: young adults without Down syndrome; Group 3: seniors without Down syndrome. SD: standard deviation.

For the results of binocular visual acuity at far and near distance, Table 2 shows the current VA with spectacles if subjects wore them and with the new prescription. Significant differences ($p < 0.001$) were found between the three groups. Group 1 got the lowest VA, while Group 2 showed the highest results.

**Table 2.** Presenting visual acuity with the new prescription at far and near distance.

| | Presenting Visual Acuity with the New Prescription | | | |
|---|---|---|---|---|
| | Far Distance | | | |
| | Presenting VA (logMAR) | Range | New VA (logMAR) | Range |
| | 0.24 ± 0.18 *** | 0.8 to −0.1 | 0.15 ± 0.10 *** | 0.8 to −0.1 |
| Group 2 | −0.16 ± 0.0 *** | 0.2 to −0.12 | −0.18 ± 0.00 *** | 0.2 to −0.2 |
| Group 3 | 0.0 ± 0.10 *** | 0.2 to −0.1 | −0.02 ± 0.00 *** | 0.2 to −0.1 |
| | Near Distance | | | |
| | Presenting VA (logMAR) | Range | New VA (logMAR) | Range |
| Group 1 | 0.23 ± 0.10 *** | 0.6 to 0.0 | 0.16 ± 0.10 *** | 0.6 to −0.05 |
| Group 2 | −0.06 ± 0.10 *** | 0.2 to −0.1 | −0.06 ± 0.10 *** | 0.2 to −0.1 |
| Group 3 | 0.01 ± 0.00 *** | 0.2 to −0.1 | 0.00 ± 0.00 *** | 0.15 to −0.1 |

VA: visual acuity. Group 1: young people with Down syndrome; Group 2: young people without Down syndrome; Group 3: seniors without Down syndrome. (*** $p < 0.001$).

With respect to refractive error, in Group 1, participants had glasses in 82.4% of cases, but 33.9% did not wear them. In Group 2, 66.2% had glasses, and 4.4% did not wear them. In Group 3, almost all the subjects, 98.2%, had and wore glasses. Statistical analysis showed significant differences between all groups ($p < 0.001$). The data presented in this section correspond to the right eyes since there were no significant differences between both eyes in all groups.

The refractive error found in each group is presented in Table 3. It shows the sphere, cylinder, spherical equivalent (SE) and range in the current prescription and in the one found after the new refraction. Significant differences ($p = 0.002$) were found between Group 2 and Group 3 in the spherical values in the prescription they wore. Regarding the amount of the cylinder, significant differences ($p < 0.001$) between Group 1 and the others were found. In Group 1, 50% of the participants had a cylinder higher than −1.00D. The astigmatism was 50% with the rule, 34.2% against the rule and 15.8% oblique. Significant differences were found between Groups 1 and 2 ($p = 0.025$) and between Groups 1 and 3 ($p = 0.008$). The spherical values and the SE in the presenting prescription show that there is no significant differences between the DS group and the others, while there is a significant difference in astigmatism values.

**Table 3.** Presenting and optimal prescription.

| Groups | Presenting Prescription | | | | |
| | Mean Sphere (D) ± SD | Range (D) | Mean Cylinder (D) ± SD | Range (D) | SE |
| --- | --- | --- | --- | --- | --- |
| Group 1 | −1.00 ± 4.23 | ±7.00; −14.00 | −1.16 ± 1.02 *** | 0.00; −3.25 | −1.33 ± 3.95 |
| Group 2 | −1.19 ± 2.59 ** | ±6.00; −11.25 | −0.38 ± 0.67 *** | 0.00; −3.50 | −1.38 ± 2.66 ** |
| Group 3 | 0.11 ± 1.36 ** | ±3.50; −4.50 | −0.57 ± 0.68 *** | 0.00; −3.00 | −0.17 ± 1.35 ** |
| | Optimal Prescription | | | | |
| | Mean Sphere (D) ± SD | Range (D) | Mean Cylinder (D) ± SD | Range (D) | SE |
| Group 1 | −1.05 ± 4.20 | ±7.00; −15.00 | −1.73 ± 1.00 *** | 0.00; −4.00 | −1.92 ± 4.20 |
| Group 2 | −0.82 ± 2.60 | ±6.00; −10.50 | −0.45 ± 0.60 *** | 0.00; −3.50 | −1.05 ± 2.70 |
| Group 3 | 0.00 ± 1.40 | ±3.50; −4.50 | −0.55 ± 0.60 *** | 0.00; −2.50 | −0.27 ± 1.40 |

Group 1: young adults with Down syndrome; Group 2: young adults without Down syndrome; Group 3: seniors without Down syndrome. D: dioptre; SD: standard deviation; SE: spherical equivalent. (** $p < 0.01$) (*** $p < 0.001$).

Considering the spherical equivalent (SE) in Group 1, myopia higher than −1.00D was present in more than 50% of the participants. There were significant differences ($p = 0.003$) between Group 2 and Group 3. The spherical values in the new prescription showed no significant differences between any of the groups ($p = 0.247$). Significant differences were found in the amount of the cylinder ($p < 0.001$) between Group 1 and the others. In Group 1, the new prescription revealed astigmatism up to 1.00D in more than 78% of the participants, which was a significant difference ($p < 0.001$) compared with the previous refractive error. In Group 1, the astigmatism was with the rule in 36.2% of participants, 33.3% against the rule and 30.5% oblique. Considering SE, the prevalence of myopia, in the optimal prescription, was higher than 50% in Group 1, 44% in Group 2 and 34% in Group 3. No significant differences were found ($p = 0.083$) between groups as shown in Table 4. But, in Group 1, differences ($p < 0.001$) were found between the current and the new prescription (SE and astigmatic error). The same happened in Group 3 for the SE ($p = 0.023$) and the astigmatic error ($p = 0.041$). In Group 2, the new prescription showed more positive values with no significant differences. Regarding the addition for near distance, there was a significant difference ($p < 0.001$) between Groups 1 and 2 with Group 3.

**Table 4.** Refractive error in the presenting and optimal prescription.

| | Presenting Prescription | | | Optimal Prescription | | |
| | Hyperopia ≥ +1.00D | Emmetropia | Myopia ≤ −1.00D | Hyperopia ≥ +1.00D | Emmetropia | Myopia ≤ −1.00D |
| --- | --- | --- | --- | --- | --- | --- |
| | n (%) | n (%) | n (%) | n (%) | n (%) | n (%) |
| Group 1 | 13 (18.8) | 35 (50.7) | 21 (30.4) | 17 (24.5) | 17 (24.6) | 35 (50.9) |
| Group 2 | 4 (5.9) | 32 (47.1) | 32 (47.0) | 17 (25) | 21 (30.9) | 30 (44.1) |
| Group 3 | 12 (21.8) | 17 (30.9) | 26 (47.3) | 15 (27.3) | 21 (38.2) | 19 (34.5) |

D: dioptre. Group 1: young adults with Down syndrome; Group 2: young adults without Down syndrome; Group 3: seniors without Down syndrome.

Strabismus was found in 64.7% of the participants in Group 1, and esodeviation was found in 55% of the cases. Nystagmus was present in 22.7% of participants. The statistical differences in both variables with the other two groups were significant ($p < 0.001$). There were no significant differences in the near point of convergence ($p = 0.439$). As was expected, Group 1 obtained very low results in stereopsis. Only 6% of the participants obtained 140 or less arcsec. There were significant differences between Group 1 and the other two ($p < 0.001$).

## 4. Discussion

The number of publications about ocular anomalies in adults with DS is very low, hence the importance of this work. Fortunately, the increased life expectancy of people with DS means that they require more medical care throughout their lives, including vision care. To our knowledge, this study is the first in which visual acuity, refractive error, accommodation and binocular vision in young adults with DS have been compared to young adults and seniors without DS. The aim was to evaluate some aspects of visual function in adults with DS and to compare them with the results obtained in young adults and seniors without DS.

As it occurs in other systems in people with DS, premature ageing may explain worse visual function. The current analysis shows that the characteristics of visual function in young adults with DS is more similar to young adults without DS than to seniors. Lower VA, higher refractive errors and strabismus are more frequent in young adults with DS, and this does not seem to be related to ageing, but to the DS condition. It has been widely reported that children with DS have more visual impairments than healthy controls without DS [4,13,31]. The same is observed in adults, but there is less information about them [21,24]. What has been published shows discrepancies in results because there are huge differences in samples and evaluation procedures.

The current study shows that more participants with DS had glasses, but 33.9% of them did not usually wear them. If the proper prescription is not worn in childhood, it will hinder the correct development of vision, and VA, binocularity or accommodation will be distorted. The noncompliance in this group is lower than what was reported by Adyanthaya and Van Splunder [28,30], with noncompliance being around 44%. The reason could be a wrong prescription [32] or the fact that DS patients are not aware of the benefits of wearing glasses [28]. In this study, the VA at far and near distance was assessed with the same test in all samples to produce consistent analyses. In young adults with DS, the VA was lower than in controls, even with the new prescription, although there was an improvement in 40% of the participants. This could be associated with the neurological anomalies that people with DS present, as mentioned before [13,21–23]. Refractive error in children with DS develops differently to the typical development, especially after two years of age [33,34]. The prevalence of myopia and hyperopia mentioned in the literature varies widely—from 4 to 60% [35–37]. In this study, refractive errors were found in 75% in Group 1. The myopia values obtained in young adults with DS were higher (50.9%) than those published by other authors [24,25,28,38], but were similar to the results obtained in this study for young adults without DS. This could be related to the kind of near activities they perform in their daily life. As it occurs in the general population, near tasks may be associated with myopic progression. Astigmatism was more common in participants with DS compared to the control group, which agrees with other authors' results. The prevalence (70%) was higher than what was found by Ljubic, ranging from 6.3% to 40.1% [24]. In addition, in more than 78% of the participants, the astigmatism values were higher than −1.00D, and in one-third of them, they were oblique. The increase in the astigmatism values with age in DS has also been mentioned in the literature [25,33]. In this study, astigmatism values higher than −1.00D were present in more than 78% of the participants with DS. Other authors have found values ranging from 66.8% to as high as 94.1% [39]. The most common orientation was oblique in one-third of the group.

Regarding the accommodative response, no significant differences were found between groups of young people with and without DS. It is acknowledged that children with DS are more likely to have accommodative deficits than children with typical development [14,40,41], and young adults with DS might be expected to have them too [41]. But in this study, lower VA at near distance in young adults with DS does not seem to be related to a lack of accommodation [42]. Accommodative response is one of the most difficult aspects of visual function to evaluate because great collaboration is needed for the assessment. In objective measurements, when subjects have to maintain focus on the target during the procedure, it

is difficult for the examiner to know if they are performing properly or getting tired. That is why the push-down method has been used.

Strabismus was observed in 64.7% of the participants in the DS group. This result is higher than the results published by Adio (9.5%), Creavin (20–40%) and Haugen (42%) [35,38,43]. Moreover, esodeviation was the most frequent deviation, which is in agreement with the previous studies [13]. It should be noted that strabismus impairs depth perception and distance calculation. The high incidence of strabismus in people with DS has a negative impact on their daily life, both in terms of mobility and in the performance of near vision tasks. That is why early detection and appropriate treatment is so important.

Nystagmus was found in 22.7% of participants in the group with DS, which was similar to Kim's article [37] and higher than Fong and Li's results [21,27]. Nystagmus is reported to affect between 2 and 33.3% of individuals with DS [39]. No significant differences were found in colour vision among the groups, and no anomalies have been found, as has been stated [44].

A limitation of this study is that, as previously explained, accommodation response was assessed using the push-down method. Thus, the Nott retinoscopy or the push-up methods should be applied to compare the results with different techniques. Moreover, the measurement of refractive error using cycloplegia has been mentioned by other authors, and if it had been used, the results could be dissimilar. Future studies that include this assessment will provide valuable information on the state of visual function in people with DS.

In some regards, people with DS need much more help because they are more vulnerable. Nowadays, they have the possibility of getting different jobs in order to achieve a certain independence and join in social life. We should point out that people with DS sometimes cannot report visual and ocular symptoms because of their communication problems. It is even possible that they feel no pain, or they do not understand that something is wrong. They should have periodic and complete examinations to ensure proper visual function and to prevent future problems before they occur. Professionals must adapt procedures individually and consider how people with DS may need more time to answer, more breaks during the evaluation or shorter sessions. This will ensure a correct diagnosis and treatment management.

This paper shows that the young people with DS studied have visual problems similar to the general population but with a higher incidence. The results obtained in this study show that more than 40% of the DS participants improved their VA with the optimal prescription. Lower VA and binocularity problems hinder their daily life, hence the importance of providing appropriate treatment. A surprising result is that accommodation in the group studied does not show any anomalies, contrary to published studies.

People with intellectual disabilities may achieve a better performance if they have a better visual function and if they enhance their quality of life. Health systems should provide vision care programmes to this special population.

**Author Contributions:** Conceptualization, Y.M.-P. and C.S.-R.; methodology, Y.M.-P., C.S.-R. and G.G.-M.; formal analysis, Y.M.-P., C.S.-R. and G.G.-M.; investigation, G.G.-M., A.L.G.-H. and Y.M.-P.; resources, Y.M.-P., C.S.-R. and G.G.-M.; writing—original draft preparation, Y.M.-P., G.G.-M. and A.L.G.-H.; writing—review and editing, Y.M.-P., C.S.-R. and V.B.-S.; supervision, Y.M.-P., C.S.-R., V.B.-S. and G.G.-M. All authors have read and agreed to the published version of the manuscript.

**Funding:** This research received no external funding.

**Institutional Review Board Statement:** The study was conducted in accordance with the guidelines of the Declaration of Helsinki and approved by the Ethics Committee of Hospital Clínico San Carlos in Madrid (Spain) (protocol code 141/359-E, approved on 24 November 2011).

**Informed Consent Statement:** Informed consent was obtained from all subjects involved in the study. If any participant had a tutor or legal guardian, they were the ones who signed the consent.

**Acknowledgments:** The authors want to thank all the participants for their collaboration, as well as their families. Special thanks to Aprocor and all the staff for the facilities to develop this study. The authors also want to thank the Statistical Department of Complutense University of Madrid.

**Conflicts of Interest:** The authors declare no conflict of interest.

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
