# Peer review of "Vision Impairments in Young Adults with Down Syndrome"

_2411-5150, 2023_

Round 1

Reviewer 1 Report

The article "Vision impairments in young adults with Down Syndrome” evaluates vision problems in young adults with DS and compares them with peer and senior groups without DS. Data showed more DS participants obtained a better visual acuity with the optimal prescription, highlighting the value of regular eye examinations for people with DS. 

(1) According to a study in REF30, older people with DS had more visual impairment than younger people. Does the inappropriate prescription affect the vision of DS patients? For example, in the DS group, is there a difference in refractive error or BCVA between individuals who wear almost optimal prescriptions and those who do not?

(2) Is the BCVA deficiency consistent with vision problems in early childhood?

(3) The authors suggest improving the visual care procedure for patients with DS and mention that "a trial was conducted to confirm that all participants understood what they had to answer". It would be helpful to have more details about how this was achieved, e.g. what procedures were adjusted? What was the timeframe of evaluation for people with DS?

Author Response

Dear Sir/Madam:

Thank you for your comments. I have changed the conclusions in the main document. The response is in the attached document. Please see the attachment.

Yours sincerely,

Reviewer 2 Report

Martin-Perez et al. realized a very interesting article describing the “Vision impairments in young adults with Down syndrome”.  This is an interesting study examining visual function in young adults with Down syndrome compared to controls without Down syndrome. The topic is important given the lack of research on vision issues in adults with Down syndrome. The study design and methods are appropriate to address the research aims. However, there are some areas that need clarification and expansion to improve the quality of the manuscript.

·      The introduction provides good background on Down syndrome and the rationale for studying vision in this population. However, more details are needed on prior research on vision function in adults with Down syndrome to contextualize the current study. What are the existing knowledge gaps?

·      In the methods, please provide more details on the recruitment process and inclusion/exclusion criteria beyond Down syndrome diagnosis and age. How was sample size determined?

·      The results are overall clearly presented, but some parts need more explanation. For example, in presenting the refractive error data, please highlight the key significant between-group differences found.

·      In the discussion, put the main findings into context of prior literature. How do your results confirm or contradict previous studies? Elaborate on the implications of the high rate of strabismus and uncorrected refractive error. Furthermore, I suggest adding data related to recent bulk transcriptomics studies which could represent a strong substrate to enforce the role of described molecular mechanisms, such as the recent PMID: 36290689, PMID: 36490268 and PMID: 32184807.

·      Discuss limitations of the study, such as the lack of cycloplegic refraction. Suggest future studies to address gaps.

·      The conclusion should highlight the key novel findings and the implications for vision care in this population.

Minor comments:

·      Abbreviate Down syndrome as DS after first use.

·      Define logMAR and other technical terms at first use.

·      Check references to make sure all cited studies are included in the reference list.

Overall, this is a worthwhile study providing useful data on an understudied topic. Addressing the above comments and expanding certain sections would enhance the quality and clarity of the manuscript.

The manuscript will benefit of a relevant English check.

Author Response

Dear Sir/Madam,

thank you very much for your suggestions. I have responded in the document attached. I would like to tell that the English language has been checked by a native speaker.

Yours sincerely,

Yolanda Martín

Round 2

Reviewer 2 Report

The authors addressed all suggested points.

The English only requires several minor revisions.